# Research on Power System Day-Ahead Generation Scheduling Method Considering Combined Operation of Wind Power and Pumped Storage Power Station

**Zhi Zhang [1,*], Dan Xu [2], Xuezhen Chan [3] and Guobin Xu [3]**

1    Department of Electrical Engineering, Tsinghua University, Beijing 100084, China
2    China Electric Power Research Institute, Beijing 100192, China
3    Hubei Provincial Key Laboratory for Operation and Control of Cascaded Hydropower Station, China Three Gorges University, Yichang 443002, China
*    Correspondence: zhangzhi6881@163.com; Tel.: +86-187-0103-3261

**Abstract:** In the proposed wind-storage combined operation technology, the storage side is foreseen to play a significant role in power system day-ahead generation scheduling. Based on the operational characteristics of pumped storage power stations, the day-ahead dispatching method of a power system with wind farms and pumped storage power stations is studied. The dispatching mode that aims at the lowest operating cost is proposed, taking into consideration the coordination relationship between the scheduling benefit of pumped storage power stations and the total peak-shaving economy of the system and the fluctuation of new energy output. First, taking the constraint of reservoir capacity, the output power, and the daily pumping power of the pumped storage power station into account, a day-ahead generation scheduling model is constructed, with the objective of minimizing costs. Then, the imperial competition algorithm is applied to the proposed model. Finally, the algorithm is compared with the standard particle swarm optimization algorithm. The simulation results based on standard 4-unit and 10-unit systems indicate that the proposed method is effective and robust for a power system with wind power and pumped storage power stations.

**Keywords:** combined operation of wind power and pumped storage power station; integration of wind power; imperial competition algorithm; unit commitment

## 1. Introduction

Wind power is a non-polluting and environmentally friendly renewable energy source, characterized over the last few decades by wide distribution, high energy density, high utilization efficiency, and suitability for large-scale development. Thus, there has been increasing global attention to the technologies of wind power generation. Due to its inherent nature of uncertainty and fluctuation, its large-scale application may challenge the traditional scheduling method [1]. Therefore, it is of great theoretical and practical significance to study the day-ahead scheduling method of power systems with high wind integration.

A great deal of previous research has been conducted on using traditional thermal power units as a regulating measure against wind power fluctuations. However, there are many limiting factors for the peak regulation of traditional thermal power units, such as diseconomy, inefficiency, and instability. It is difficult to realize lower operating costs and ideal stabilization effects by merely using the remaining wind power in combined operation. It has been shown that large-scale power storage technology is expected to offer a potential solution. Correspondingly, how to foster the large scale application of wind power by combining it with pumped storage power stations has become a crucial issue. Ref. [2] studied the optimal capacity ratio of the pumped storage power station under the premise of known wind power output. Heterogeneous wind power–hydropower joint optimization models with the goal of maximizing economic benefits have been established

in References [3–5]. In order to accommodate as much wind energy as possible, Ref. [6] made a simulation to verify the efficacy of the wind power–pumped storage combined operation scheme. References [7,8] evaluated the effect of combined wind power and pumped storage from four aspects: power flow distribution, static stability, wind power penetration, and economic benefit. Ref. [9] proposed a wind-storage optimal scheduling model that considered the electricity market framework based on the day-ahead plan, attending to the demand for and price of electricity, with the goal of minimizing the costs caused by its insufficient delivery. References [10,11] introduced the Monte Carlo method and the MGSO-ACL algorithm, respectively, to build a capacity-allocation planning model combining wind farm and pumped storage power stations that explored the rational allocation of wind and photovoltaic storage capacity. Ref. [12] proposed the constraints of pumped storage power stations by considering hydraulic constraints, and jointly optimized the scheduling strategy of pumped storage power stations and the unit combination strategy of conventional power plants under the deterministic-unit combination and random-unit combination models.

At present, research on the optimal operation of pumped storage power stations mainly focuses on the combined power supply of integrating other energy sources to facilitate grid scheduling. Focusing on the combined power supply limits the pumped storage to the power generation side and does not take full advantage of the flexibility of pumped storage resources. Thus, there is still space to develop the utilization of pumped storage. Moreover, few studies have considered the scheduling benefit of pumped storage power stations caused by the loss of pumping efficiency, and previous research has failed to consider the coordination relationship between the scheduling benefit of pumped storage power stations and the total peak-shaving economy of the system and the fluctuation of new energy output.

Therefore, a wind power–hydropower joint day-ahead dispatching model, which considers the peak-regulating characteristics of pumped storage, is constructed in this paper. In particular, we propose a notion that regards the pumped storage power station as a technical method for mitigating the impacts of wind power's fluctuating output. Finally, the imperial competition algorithm is used to solve the model. We tested the performance of the proposed scheduling method using a case study based on the IEEE 4- and IEEE 10-machine standard example, which is verified as having excellent optimization and robust performance for power systems with wind farms and pumped storage power stations.

## 2. Dynamic Economic Scheduling Model of Power System with Wind Farm and Pumped Storage Power Station

The economic scheduling of a power system involves researching and formulating the generation plan of each unit over a certain time period to minimize the total cost of power generation, which needs to satisfy the system and unit constraints. The conventional units are subject to system operation, security constraints, and the characteristics of the unit itself. Therefore, the economic scheduling of a power system is essentially a very complex mathematical optimization problem [13].

### 2.1. Objective Function

In the actual power system, the wind farm has a low operating cost compared with a thermal power plant [14–20], so it can be ignored, but the switching cost of the unit in the start–stop stage should be taken into account [21–24]. Because the fuel characteristics of the conventional unit are affected by the load level, the power generation efficiency will be at its highest when the output of the conventional unit is kept near the rated output power. Therefore, the objective function of the model includes the fuel cost and the start–stop cost of the unit.

The objective function can be formulated as follows:

$$f\left(P_n^h\right) = \sum_{t=1}^{T}\sum_{n=1}^{N}\left[C_n\left(P_n^t\right)*u_{nt}+S_{nt}*u_{nt}\left(1-u_{n(t-1)}\right)\right] \tag{1}$$

where $T$ is the system scheduling period; $N$ is the number of generators in the system; $P_n^t$ denotes the output of unit $n$ at time $t$; $C_n$ is the operating cost coefficient of unit $n$; $u_{nt}$ is the working state of unit $n$ in period $t$, $u_{nt}=0$ indicates that the unit is in shutdown state, and $u_{nt}=1$ indicates that is in startup state; $S_{nt}$ is the startup cost of unit $n$ at time $t$.

*2.2. Constraint Condition*

2.2.1. System Power Balance Constraint (Ignoring Network Loss):

The system power balance constraint can be formulated as follows:

$$\sum_{n=1}^{N}P_n^t*u_{nt} + P_w^t = P_{LD}^t \tag{2}$$

where $P_w^t$ is the predicted value of wind power in $t$ period; $P_{LD}^t$ is the load forecast value of the $t$ period.

2.2.2. Conventional Unit Output Constraints

The conventional unit output constraints can be formulated as follows:

$$P_n^{\min} \leq P_n^t \leq P_n^{\max} \tag{3}$$

where $P_n^{\max}$ and $P_n^{\min}$ denote the maximum and minimum output of unit $n$.

2.2.3. System Spinning Reserve Capacity

Positive spinning reserve constraint:

$$\sum_{n=1}^{N}R_{u,n}^t \geq \Delta P_{LD}^t + u_s\%P_w^t \tag{4}$$

$$R_{u,n}^t = \min\left(P_{n,\max}-P_n^t, U_{Rn}\Delta T\right) \tag{5}$$

Negative spinning reserve constraint:

$$\sum_{n=1}^{N}R_{d,n}^t \geq d_s\%P_w^t \tag{6}$$

$$R_{d,n}^t = min\left(P_n^t - P_{n,\min}, D_{Rn}\Delta T\right) \tag{7}$$

where $d_s\%$ and $u_s\%$ represent the demand coefficient for reserve capacity when the wind power output is underestimated and overestimated, $\Delta P_{LD}^t$ is the reserve capacity at time period $t$, which is used to cope with unit outage and load forecasting errors; $R_{u,n}^t$ and $R_{d,n}^t$ represent the negative spinning reserve capacity and positive spinning reserve capacity of unit $n$ in period $t$ respectively, and their values are usually related to the system load.

2.2.4. Unit Climbing Rate Constraint

Uphill climbing rate constraint:

$$P_n^t u_{nt} - P_n^{t-1}u_{n(t-1)} \leq U_{Rn}\Delta T \tag{8}$$

Downhill climbing rate constraint:

$$P_n^{t-1} u_{n(t-1)} - P_n^t u_{nt} \leq D_{Rn} \Delta T \tag{9}$$

where $D_{Rn}$ is the downhill climbing rate of unit $n$. Instead, $U_{Rn}$ is the uphill climbing rate of unit $n$, their unit is MW/h; $\Delta T$ is a unit scheduling period, typically taking values of 1 h.

### 2.2.5. Conventional Unit Start–Stop Time Constraints

The conventional unit start–stop time constraints can be formulated as follows:

$$\left( T_n^t - T_{u,n} \right) \left( u_{n(t-1)} - u_{nt} \right) \geq 0 \tag{10}$$

$$\left( -T_n^t - T_{d,n} \right) \left( u_{nt} - u_{n(t-1)} \right) \geq 0 \tag{11}$$

$$i = 1, 2, \ldots, T, n = 1, 2, \ldots, N$$

where $T_n^t$ is the number of time periods that the unit $n$ has been running in $t$ period ($T_n^t > 0$) or the number of time periods that have been shut down ($T_n^t < 0$); $T_{d,n}$ and $T_{u,n}$ denote the maximum number of continuous operation periods and minimum number of continuous outage periods of unit $n$.

### 2.2.6. Constraints of Pumped Storage Power Stations

The water volume and installed capacity of upper and lower reservoirs are the main factors affecting the input power (pumping power) and output power (generating power) of pumped storage power stations [25–30]. The output power can be quickly adjusted within the maximum power generation output of the turbine and the maximum pumping load of the pump when the storage capacity allows [31–38].

(1)    Capacity constraints of pumped storage power stations

For any period of $\tau \in T$, the constraints can be expressed as follows:

$$\frac{W_o - W_{\max}}{\eta_s} \leq \frac{\sum\limits_{t=1}^{\tau} P_G^t \eta_G}{\eta_s} - \sum\limits_{t=1}^{\tau} P_S K_S^t \leq \frac{W_o - W_{\min}}{\eta_s} \tag{12}$$

where $W_o$ is the initial water volume of the upper reservoir of a pumped storage power station; $W_{\max}$ and $W_{\min}$ represent the maximum and minimum water volume of the upper reservoir of a pumped storage power station, respectively; $P_G^t$ is the generation power of a pumped storage power station in period $t$; $P_S$ is the pumped power for a pumped storage power station; $K_S^t$ is the number of units working in the pumping condition for the time period $t$ of a pumped storage power station; $\eta_G$ is the average power conversion coefficient; $\eta_S$ is the average water conversion coefficient.

(2)    Output power constraint of pumped storage power station

The output power of pumped storage power stations must satisfy the upper and lower demands of its constraints and avoid some units working in power-generation conditions while other units work in pumping conditions.

$$\begin{cases} K_G^t P_{G,min} \leq P_G^t \leq K_G^t P_h \\ K_S^t K_G^t = 0 \\ K_S^t + K_G^t \leq K \end{cases} \tag{13}$$

where $K_G^t$ is the number of units working in power-generation conditions in a pumped storage power station in time period $t$; $K$ is the total number of units in a pumped storage

power station; $P_h$ is the rated power of generating units for a pumped storage power station; $P_{G,min}$ is the minimum output of generating units for a pumped storage power station.

(3)　Daily extraction power constraint of pumped storage power station

$$\frac{\sum\limits_{\tau \in T} P_G^t \eta_G}{\eta_S} = \sum_{\tau \in T} P_S K_S^t \tag{14}$$

## 3. Imperial Competition Algorithm

The imperial competition algorithm is an intelligent algorithm with global search ability. It draws on the process by which feudal empires invaded each other's colonies and developed and grew through mutual competition.. Thereby, the initial population is defined as a country and, according to degrees of power, the two countries are categorized into an 'imperialist country' and a 'colonial country' [16]. Power is, here, an indicator of whether a country is strong or not, which is related to the objective function of the model. The optimal solution is obtained by simulating the process of competition between empires as they acquire colonies. The algorithm can be partitioned into four parts: initializing empire, annexing colonial countries, competition between imperialist countries, and the demise of the weakest empire.

### 3.1. Initializing Empire

A g-dimensional decision variable, which is defined as a country, is generated in the space that needs to be searched. While its location is randomly distributed in the search space and is defined as country = $[x_1, x_2, \dots, x_g]$, the function value is $f_{country}$, and the power of the country $n$ is defined as:

$$P_n = \left| \frac{f_n - \max\limits_{m}\{f_i\}}{\sum\limits_{i=1}^{m} (f_n - \max\limits_{m}\{f_i\})} \right| \tag{15}$$

$N_c$ countries with greater power are designated as imperialist, while those with less power are designated as colonial. Colonial countries are taken over according to the size of imperialist powers; the greater the power, the more colonial countries. An imperialist country and its colonial countries form an empire.

### 3.2. Annexing Colonial Countries

The position in the search space represented by the colonial state is close to the position represented by the imperialist state, randomly moving a certain distance [14]. Let the moving distance of the colonial country be $l$:

$$l \sim U(0, \delta \times l_D) \tag{16}$$

where $l_D$ is the straight line's distance between the colonial country and its imperialist location and $\delta > 1$.

Assuming that the angle between the moving direction of the colonial country and the offset of its connection with the position of the imperialist country is $\theta$, then:

$$\theta \sim U(-\psi, \psi) \tag{17}$$

where $\psi$ is the adjustment parameter of the offset angle.

It is noteworthy that when an imperialist country annexes a colonial country, a change in the spatial location of the colonial country may result in a situation in which it is more powerful than the imperialist country to which it belongs, thus potentially replacing it as the new imperialist country of the empire.

The process of imperialist countries annexing colonial countries is shown in Figure 1.

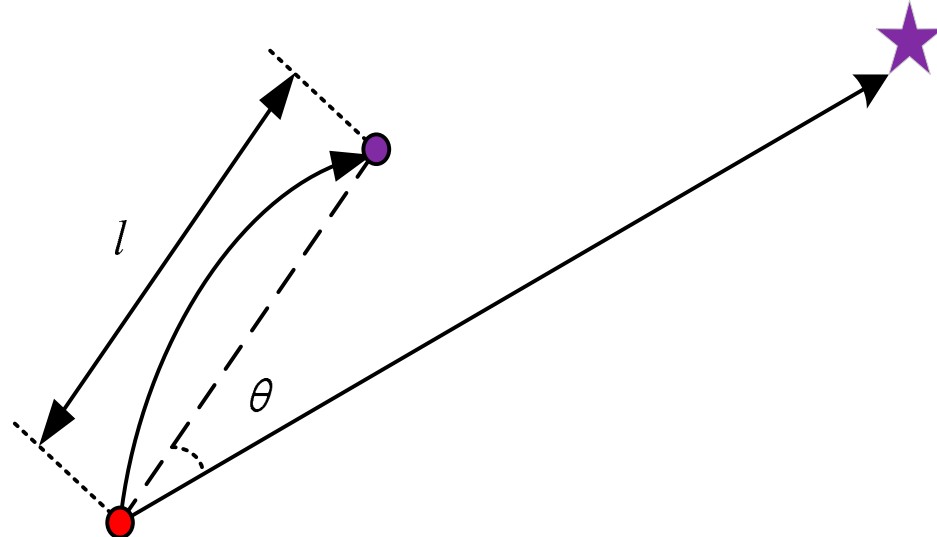

**Figure 1.** Process of Imperialist Countries Annexing Colonial Countries.

*3.3. Competition between Imperialist Countries*

Defining the total power of the empire as:

$$C_{Ar} = \left| (f_r + \sigma \times \overline{f}_{rcol}) - \max_{N_C}\left\{ f_i + \sigma \times \overline{f}_{rcol} \right\} \right| \tag{18}$$

where $f_r$ is the objective function value of imperialist country $r$, $\sigma$ is the weight parameter, $\overline{f}_{rcol}$ is the average value of the objective function for possession of a colony for imperialist country $r$.

The result of competition among imperialist countries is to select colonial countries from the empires with the weakest total power, then allocate them to the other $N_c - 1$ empires with a certain probability [14], the probability that the empire $j$ will possess it is:

$$P_j = \left| \frac{C_{Aj}}{\sum\limits_{i=1}^{N_C-1} C_{Aj}} \right| \tag{19}$$

*3.4. The Demise of the Weakest Empire*

After the competition between imperialist countries, the empire which loses all its colonial countries will be extinct. After a certain period of time, if there is a most powerful empire in the search space that also contains purely one imperialist country and all colonial countries, so the ideal situation is obtained. That aforesaid imperialist country is the optimal solution, and the algorithm terminates. Otherwise, return to Section 3.2.

The detailed steps of the imperialist competitive algorithm are shown in Figure 2 and Appendix A.

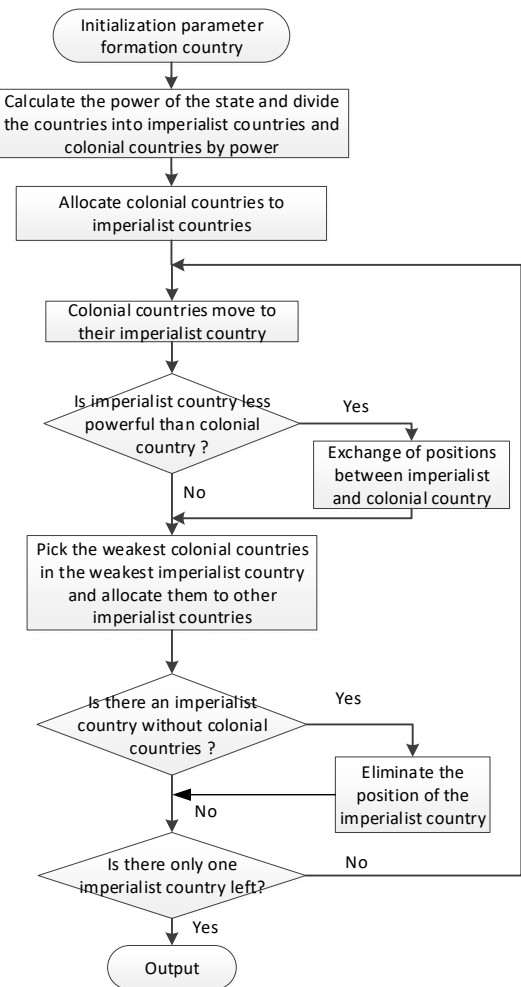

**Figure 2.** Algorithm flow chart.

## 4. Example Simulation and Result Analysis

### 4.1. Simulation Based on 4 Machines and 6 Nodes

The simulation example includes three conventional generator sets, one wind farm, five lines, one inverter, and two transformers with transformation taps. The system wiring diagram is illustrated in Figure 3. Subsequently, the pumping and storage process at a pumped storage power station is equivalent to load change. The standby demand for positive rotation of conventional units is 20 MW, and the standby demand for negative rotation is 2% of the minimum load of the system. Parameters of conventional units and lines can be discovered in Ref. [39].

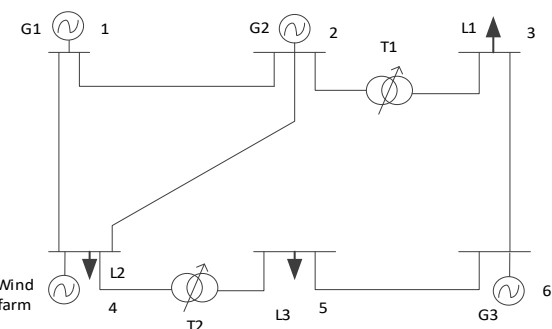

**Figure 3.** 4-machine and 6-node system wiring diagram.

The daily load forecast data is depicted in Table 1. The percentages of L1, L2, and L3 in the total load are 20%, 40%, and 40%, respectively. The wind farm is connected at Node 4, and its predicted output is shown in Table 1. Assuming that the wind turbine does not provide rotating standby and does not consider the possibility of forced outage, the relevant parameters of the wind turbine are $v_{in}$ = 3 m/s, $v_{0ut}$ = 25 m/s, and $v_R$ = 15 m/s, and the standard deviation of wind speed prediction error is 0.5.

**Table 1.** Forecast data of daily load and wind power output.

| Time | $P_L$ (MW) | $Q_L$ (MW) | $P_W$ (MW) | Time | $P_L$ (MW) | $Q_L$ (MW) | $P_W$ (MW) |
|------|-----------|-----------|-----------|------|-----------|-----------|-----------|
| 1 | 209.19 | 50.4 | 51 | 13 | 324.18 | 69.6 | 81 |
| 2 | 224.35 | 45.8 | 69 | 14 | 314.60 | 70 | 80 |
| 3 | 221.67 | 45.6 | 80 | 15 | 326.86 | 71.6 | 77 |
| 4 | 236.73 | 42.5 | 79 | 16 | 285.25 | 73.5 | 32 |
| 5 | 226.12 | 44.6 | 81 | 17 | 260.00 | 71.6 | 4 |
| 6 | 244.48 | 48.2 | 84 | 18 | 237.52 | 70.9 | 10 |
| 7 | 273.39 | 49.9 | 98 | 19 | 255.97 | 70.7 | 9 |
| 8 | 285.31 | 51.1 | 95 | 20 | 239.11 | 68.2 | 5 |
| 9 | 283.56 | 50.9 | 68 | 21 | 243.31 | 67.1 | 6 |
| 10 | 276.25 | 59.5 | 61 | 22 | 282.69 | 66.9 | 56 |
| 11 | 328.61 | 67.1 | 100 | 23 | 281.25 | 55.1 | 74 |
| 12 | 317.59 | 67.9 | 86 | 24 | 248.75 | 56.2 | 52 |

In order to verify the effectiveness of the model and algorithm proposed in this paper, two modes are designed to simulate the examples.

Mode 1: Economic dispatching of wind power system without considering pumped storage.

Mode 2: Economic dispatching of wind power system with consideration of pumped storage.

The output of grid-connected conventional generator units is illustrated in Table 2.

**Table 2.** Output of grid-connected conventional units.

| Time | Unit Output Arrangement in Mode 1 (MW) | | | Unit Output Arrangement in Mode 2 (MW) | | |
|------|------|------|------|------|------|------|
| | G1 | G2 | G3 | G1 | G2 | G3 |
| 1 | 158 | 0 | 0 | 158 | 0 | 0 |
| 2 | 155 | 0 | 0 | 155 | 0 | 0 |
| 3 | 142 | 0 | 0 | 142 | 0 | 0 |
| 4 | 158 | 0 | 0 | 158 | 0 | 0 |
| 5 | 145 | 0 | 0 | 145 | 0 | 0 |
| 6 | 160 | 0 | 0 | 160 | 0 | 0 |
| 7 | 175 | 0 | 0 | 175 | 0 | 0 |
| 8 | 190 | 0 | 0 | 190 | 0 | 0 |
| 9 | 216 | 0 | 0 | 216 | 0 | 0 |
| 10 | 215 | 0 | 0 | 215 | 0 | 0 |
| 11 | 219 | 0 | 10 | 219 | 0 | 10 |
| 12 | 216 | 0 | 16 | 216 | 0 | 16 |
| 13 | 221 | 10 | 12 | 221 | 10 | 12 |
| 14 | 210 | 10 | 15 | 211 | 10 | 14 |
| 15 | 221 | 10 | 19 | 220 | 10 | 20 |
| 16 | 217 | 16 | 20 | 220 | 16 | 17 |
| 17 | 220 | 16 | 20 | 220 | 16 | 20 |
| 18 | 208 | 10 | 10 | 208 | 10 | 10 |
| 19 | 221 | 10 | 16 | 220 | 10 | 17 |
| 20 | 222 | 12 | 0 | 220 | 14 | 0 |
| 21 | 220 | 17 | 0 | 220 | 17 | 0 |
| 22 | 217 | 10 | 0 | 217 | 10 | 0 |
| 23 | 197 | 10 | 0 | 191 | 16 | 0 |
| 24 | 187 | 10 | 0 | 187 | 10 | 0 |

Table 2 clarifies that Mode 1 generator G1 is always in the startup operation state, and G2 and G3 are only put into operation during peak load hours and part of the time. Moreover, compared with Mode 1, the startup and shutdown status of conventional units in Mode 2 has not altered.

The time when the line power flow exceeds the limit and the corresponding power flow are summarized in Table 3.

**Table 3.** Line power flow out of limit.

| Time | Mode 1 | | Mode 2 | |
|---|---|---|---|---|
| | **Line 4–5 (MW)** | **Out of Limit Rate (%)** | **Line 4–5 (MW)** | **Out of Limit Rate (%)** |
| 11 | 121.4129 | 11.2 | 102.114 | 4.2 |
| 12 | 101.4247 | 7.9 | 101.1469 | 2.1 |
| 13 | 102.7312 | 5.1 | 100.2587 | 0.9 |
| 14 | 108.2157 | 4.2 | 98.1012 | 0 |
| 15 | 104.4578 | 3.9 | 97.3145 | 0 |

As can be observed from Table 3, after the pumped storage is included in the dispatching system, only Line 4–5 of all lines still has the power flow out of limit, but the out-of-limit period is reduced from 5 h in Mode 1 to 3 h in Mode 2, and the number of out-of-limit lines is also significantly curtailed. The costs of the two operation modes are listed in Table 4.

**Table 4.** Cost of system operation.

| | **Operation Cost ($)** | **Start and Stop Cost ($)** | **All-In Cost ($)** |
|---|---|---|---|
| Mode 1 | 78,705.03 | 300.00 | 79,005.03 |
| Mode 2 | 77,649.12 | 300.00 | 77,949.12 |

We can draw a conclusion from Table 4 that, although the system operation cost of Mode 1 is small, the sum of the out-of-limit quantities is the largest, indicating that this dispatching mode fails to consider the security of the system operation, resulting in the inferior applicability of dispatching decisions. Compared with Mode 1, the cost of Mode 2 shows that taking pumped storage as a means of peak shaving into the system dispatching model can mitigate the power generation cost of the system and alleviate the situation so that the power flow at the transmission section exceeds the limit. We can, therefore, be aware that fully utilizing the peak-shaving benefits of pumped storage can not only build up the economy of system operation but also have a positive impact on the safety and reliability of system operation.

### 4.2. Simulation Based on 10 Machines and 39 Nodes

The simulation example includes 10 conventional units, a pumped storage power station, and a wind farm. Ref. [15] provides conventional unit parameters and load data. Ref. [17] provides wind farm processing data, and the pumping process of pumped storage power station is equivalent to load change.

The calculation result of minimum power generation cost is the final result. Table 5a shows the power generation cost, start–stop cost, unit status, output arrangement, and rotating reserve capacity of 10 conventional generating units in 24 h after the system is added to the wind farm. The power generation cost is $511,249, the start–stop cost is $5547.4, and the total power generation cost is $516,796.4.

**Table 5.** Optimization results of a 10-machine example under imperialist competitive algorithm and power generation total cost and cpu time comparison. (**a**) Optimization results of a 10-machine example under imperialist competitive algorithm; (**b**) Power generation cost and cpu time comparison.

(a)

| Time | Generation Cost × 10⁴ $ | Start–Stop Cost × 10³ $ | Unit State | Unit Output Arrangement (MW) | | | | | | | | | | Spinning Reserve (MW) | |
|---|---|---|---|---|---|---|---|---|---|---|---|---|---|---|---|
| | | | | 1 | 2 | 3 | 4 | 5 | 6 | 7 | 8 | 9 | 10 | Usr | Dsr |
| 1 | 1.2431 | 0 | 1,100,000,000 | 455 | 173 | 0 | 0 | 0 | 0 | 0 | 0 | 0 | 0 | 282 | 63 |
| 2 | 1.2709 | 0 | 1,100,000,000 | 455 | 189 | 0 | 0 | 0 | 0 | 0 | 0 | 0 | 0 | 266 | 79 |
| 3 | 1.4328 | 0 | 1,100,000,000 | 455 | 282 | 0 | 0 | 0 | 0 | 0 | 0 | 0 | 0 | 173 | 172 |
| 4 | 1.6757 | 0 | 1,100,100,000 | 455 | 367 | 0 | 0 | 25 | 0 | 0 | 0 | 0 | 0 | 225 | 257 |
| 5 | 1.7002 | 1.79 | 1,100,100,000 | 455 | 381 | 0 | 0 | 25 | 0 | 0 | 0 | 0 | 0 | 211 | 271 |
| 6 | 2.0465 | 0 | 1,111,100,000 | 455 | 424 | 40 | 40 | 25 | 0 | 0 | 0 | 0 | 0 | 168 | 354 |
| 7 | 2.1179 | 2.2125 | 1,111,100,000 | 455 | 388 | 80 | 80 | 25 | 0 | 0 | 0 | 0 | 0 | 204 | 358 |
| 8 | 2.3709 | 0 | 1,111,100,000 | 455 | 455 | 120 | 120 | 25 | 0 | 0 | 0 | 0 | 0 | 200 | 362 |
| 9 | 2.4692 | 0 | 1,111,100,000 | 455 | 455 | 130 | 130 | 57 | 0 | 0 | 0 | 0 | 0 | 165 | 457 |
| 10 | 2.9179 | 0 | 1,111,111,100 | 455 | 455 | 130 | 130 | 134 | 20 | 25 | 10 | 0 | 0 | 193 | 534 |
| 11 | 2.9944 | 0.9189 | 1,111,111,100 | 455 | 455 | 130 | 130 | 162 | 28 | 25 | 10 | 0 | 0 | 157 | 570 |
| 12 | 3.2562 | 0 | 1,111,111,111 | 455 | 455 | 130 | 130 | 162 | 60 | 25 | 10 | 10 | 10 | 215 | 602 |
| 13 | 2.7860 | 0.1200 | 1,111,111,000 | 455 | 455 | 120 | 130 | 123 | 20 | 25 | 0 | 0 | 0 | 169 | 513 |
| 14 | 2.4397 | 0 | 1,111,110,000 | 455 | 455 | 80 | 130 | 44 | 20 | 0 | 0 | 0 | 0 | 218 | 404 |
| 15 | 2.2886 | 0 | 1,111,100,000 | 455 | 455 | 40 | 130 | 43 | 0 | 0 | 0 | 0 | 0 | 159 | 403 |
| 16 | 1.9775 | 0 | 1,101,100,000 | 455 | 376 | 0 | 130 | 25 | 0 | 0 | 0 | 0 | 0 | 216 | 306 |
| 17 | 1.8168 | 0 | 1,101,100,000 | 455 | 284 | 0 | 130 | 25 | 0 | 0 | 0 | 0 | 0 | 308 | 214 |
| 18 | 1.9285 | 0 | 1,101,100,000 | 455 | 348 | 0 | 130 | 25 | 0 | 0 | 0 | 0 | 0 | 244 | 278 |
| 19 | 2.1141 | 0 | 1,101,100,000 | 455 | 454 | 0 | 130 | 25 | 0 | 0 | 0 | 0 | 0 | 198 | 384 |
| 20 | 2.8129 | 0.5060 | 1,101,110,111 | 455 | 455 | 0 | 130 | 162 | 44 | 0 | 10 | 10 | 10 | 171 | 546 |
| 21 | 2.4344 | 0 | 1,101,110,100 | 455 | 455 | 0 | 120 | 105 | 20 | 0 | 10 | 0 | 0 | 172 | 455 |
| 22 | 1.9663 | 0 | 1,101,100,000 | 455 | 418 | 0 | 80 | 25 | 0 | 0 | 0 | 0 | 0 | 214 | 308 |
| 23 | 1.6212 | 0 | 1,101,000,000 | 455 | 313 | 0 | 40 | 0 | 0 | 0 | 0 | 0 | 0 | 182 | 203 |
| 24 | 1.4432 | 0 | 1,100,000,000 | 455 | 288 | 0 | 0 | 0 | 0 | 0 | 0 | 0 | 0 | 167 | 178 |

(b)

| Algorithm | Operation Cost ($) | Start and Stop Cost ($) | CPU Time (s) |
|---|---|---|---|
| Standard particle swarm optimization (PSO) | 514,755 | 6550.4 | 78.3 |
| Imperial competition algorithm (ICA) | 511,249 | 5547.4 | 60.2 |

Table 5b is plotted to compare the total cost of power generation and CPU time calculated by this algorithm and the standard particle swarm optimization algorithm.

It can be seen from Table 5b that the scheduling plan obtained by the imperial competition algorithm can effectively reduce the total cost of power generation. Compared with the standard particle swarm optimization algorithm, the total cost of power generation is reduced by $4508.9, and the optimization result is better. While its CPU time is shorter, 18.1 is less than the standard particle swarm algorithm.

Figure 4 is the unit output in this example. It can be seen from Figure 4 that the output of the system unit can meet the load demand at any time in the scheduling cycle without a power shortage. The pumped storage power station has been in a state of power generation from 08:00~16:00, which is due to the high system load during this period. In order to make up for the fluctuation of wind power and ensure the stability of the system's power supply, the pumped storage unit needs to be in a state of discharge and able to adjust the power generation output at any time. The pumped storage power station is in the state of electricity at 01:00~07:00 and 17:00~22:00, mainly because the power station needs to be in the state of electricity in order to absorb more wind power at this time. Therefore, the scheduling method based on the imperialist competitive algorithm proposed in this paper is correct and effective in solving UC problems with wind farms and pumped storage power stations.

Figure 5 is a comparison of the iterative convergence performance between the imperialist competitive algorithm and the standard particle swarm algorithm. The former converges at the 20th iteration, while the latter converges at the 23rd iteration. Therefore, compared with the latter, it is obvious that the former has faster convergence speed and stronger optimization performance.

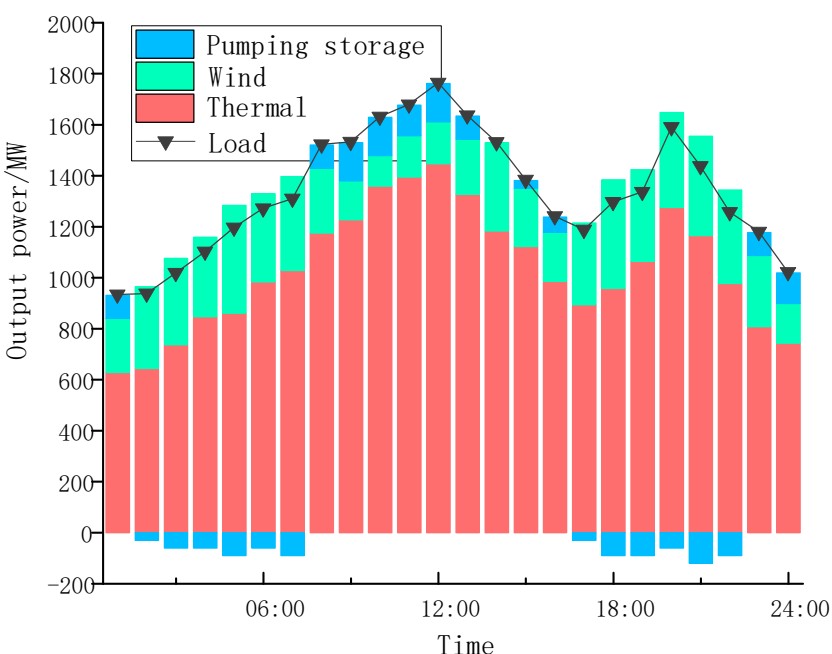

**Figure 4.** Unit output situation.

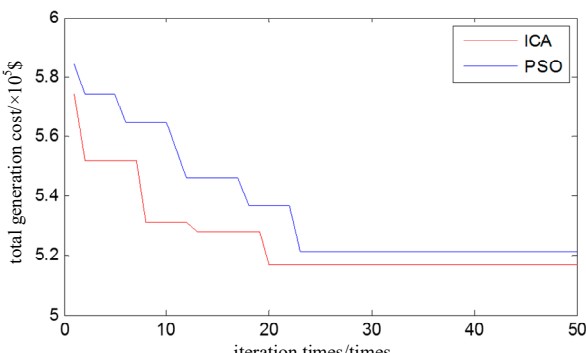

**Figure 5.** Comparison of iterative convergence curves.

## 5. Conclusions

We combined the operation characteristics of pumped storage power stations, and proposed a day-ahead scheduling method for power systems using wind-storage combined operation. The aforesaid method considers the storage capacity constraints, output power constraints, and daily pumping power constraints of pumped storage power stations. The model, which takes into account the combined operation of wind and storage, aims to reduce system operating costs. Finally, the imperialist competitive algorithm is used to solve the model. Through the simulation of the standard IEEE-4 and IEEE-10 machine system, the comparative simulation results show that the proposed method has good effectiveness and robustness in solving the dynamic economic dispatch problem of power systems that use wind power. However, the number of units considered in this paper is only 10, which cannot be applied to a larger power system. Further improvements are needed to make it suitable for larger and more complex power systems.

**Author Contributions:** Z.Z. and D.X. put forward the main research points; X.C. and G.X. completed manuscript writing and revision; Z.Z. and X.C. completed simulation research; D.X. and G.X. collected relevant background information; D.X., X.C. and G.X. revised grammar and expression. All authors have read and agreed to the published version of the manuscript.

**Funding:** This paper was funded by the technology project of state grid corporation headquarters (5100-202155294A-0-0-00).

**Institutional Review Board Statement:** Not applicable.

**Informed Consent Statement:** Not applicable.

**Data Availability Statement:** No new data created.

**Conflicts of Interest:** The authors declare no conflict of interest.

## Appendix A

| **Algorithm A1:** Imperial competition algorithm. |
|---|

| |
|---|
| Require: Initialization $\quad\triangleright$Hourly prediction data of $P_w^t$, $P_{LD}^t$, $P_G^t$, $P_S$, $K_S^t$ and $K_G^t$. |
|   **for** t = 0:24 **do** |
|   k = 1; |
|     **while** k $\leq N_{pop}$ **do** $\qquad\qquad\triangleright N_{pop}$: the number of population |
|                             $\triangleright$ Initialization |
|     **Select** some random points on dependent/independent variables |
|     **Create** the imperialist states in a 1×g matrix by |
|                   country = [x1,x2, . . . ,xg] |
|     **Evaluate** the country cost (Equation (1)) |
|       **End while** $\qquad\qquad\qquad\triangleright$End initialization |
|     **Sort** the countries based on their objective function values |
|     **Divide** colonies among imperialist |
|                 (Equations (18) and (19)) |
| *decade = 1* |
| **while** decade $\leq$ max decade **do** |
|   **Select** the ith empire. $\qquad\qquad\triangleright$Assimilation |
|   **while** all empires selected **do** |
|     **Select** the jth colony from the ith empire |
|     **while** all the colonies of ith empire selected **do** |
|       **Move** the colonies toward their relevant imperialist |
|       **Move** the jth colony toward its imperialist |
|       **if** the balance constraint was not held **then** |
|         **Reestablish** power balance |
|       **end if** |
|       **Evaluate** the jth colony (Equation (1)) |
|     **end while** |
|   **end while** $\qquad\qquad\qquad\triangleright$End assimilation |
| **Select** the ith empire $\qquad\qquad\qquad\triangleright$Revolution |
| **while** all the empires selected **do** |
|   **Select** the jth colony from the ith empire |
|   **while** all the colonies of the ith empire selected **do** |
|     **Create** random number ($P_{\text{revolution}}$) |
|     **if** $v \leq P_{\text{revolution}}$ **then** $\triangleright$ *v*: revolution rate |
|       **Select** some random points on dependent/independent variables |
|         **Create** the imperialist states in a 1×g matrix $\triangleright$country = [x1, x2, . . . ,xg] |
|         **Evaluate** the jth colony (Equation (1)) |
|     **end if** |
|   **end while** |
| **end while** $\qquad\qquad\qquad\qquad\triangleright$End revolution |
| **Select** ith empire |
| **while** just one empire will remain **do** |
|   **if** there is a colony in an empire which has lower cost than the imperialist **then** |
|     **Exchange** the positions of that colony and the imperialist |
|     **Unit** the similar empires |
|   **end if** |
| **end while** |
|   **Compute** the total cost of all empires by |
|       $C_{t,Tot}^{Imp,n} = C_t^{Imp,n} + \xi \times mean\{C_t^{Col,n}, n\}$ $\qquad\qquad\qquad\qquad\qquad\qquad$ (A1) |
|   **Eliminate** the powerless empires $\qquad\qquad\triangleright$Imperialist competition |
|   **if** *the weakest empire has a colony* **then** |
|     **Pick** the colony and give it to one of empires by the roulette wheel |
|       (Equations (18) and (19)) |
|     **else** |
|     **Allocate** the weak empire to this one by the roulette wheel (Equations (19) and (A1)) |
|   **end if** $\qquad\qquad\qquad\qquad\triangleright$End imperialist competition |
| **end while** |
| **Return** the best empire |
| **end for** |

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
