# Peer review of "Research on Power System Day-Ahead Generation Scheduling Method Considering Combined Operation of Wind Power and Pumped Storage Power Station"

_sustainability, doi:10.3390/su15076208_

Round 1

Reviewer 1 Report

The language of the manuscript is good, however, there are a few grammatical and typo errors to be corrected.

Your literature review is short. There are many novel papers in recent years that have not been studied, meaning that the literature review is not completed.

Identify research gaps leading to the need for the present study.

Novelty should be demonstrated in the introduction. What does the international reader of the journal learn from the paper? What is the research question that the paper tries to answer? What is the authors’ contribution? What is new in the methods?

Author Response

Comments to the Author

Response: The authors would like to thank the Reviewer for reviewing our paper. Based on the Reviewer’s comments, this paper is thoroughly revised and improved. The specific responses to individual comments are detailed as follows.

Comment 1:Your literature review is short. There are many novel papers in recent years that have not been studied, meaning that the literature review is not completed. Identify research gaps leading to the need for the present study.

Response: The authors agree with the Reviewer on this.

As suggested, we have further reviewed the recent works and added novel papers in recent years. Please refer to literature review of the revised manuscript.

The corresponding revisions are marked in red in Section 1 of the revised manuscript.

Comment 2:Novelty should be demonstrated in the introduction. What does the international reader of the journal learn from the paper? What is the research question that the paper tries to answer? What is the authors’ contribution? What is new in the methods?

Response: The authors would like to thank the Reviewer for this comment.

At present, the research on the optimal operation of pumped storage power stations mainly focuses on the combined power supply with other energy sources to accept grid scheduling. The combined power supply limits the pumped storage to the power generation side, which cannot give full play to the flexibility of the pumped storage resources, and the utility of the pumped storage still has a large space for utilization. At the same time, few studies have considered the scheduling benefit caused by the loss of pumping efficiency of pumped storage power stations, and failed to consider the coordination relationship between the scheduling benefit of pumped storage power stations and the total peak shaving economy of the system and the fluctuation of new energy output. Different from existing review articles, this paper constructed the wind power-hydropower joint day-ahead dispatching model with considering the peak regulation characteristics of pumped storage. In particular, we propose a notion that regards the pumped storage power station as a technical method for mitigating the impacts of the fluctuation of wind power output.

The above revisions are marked in red in Section I of the revised manuscript.

Reviewer 2 Report

This paper has carried out the research on power system day-ahead generation scheduling 2 method considering combined operation of wind power and 3 pumped storage power station. The paper covers an exciting research object, and the paper is technically sound. However, this reviewer is curious that as there any realistic/commercial 4-machine and 6-node system, where the proposed model can be applied. Additionally, this reviewer would recommend the application of the proposed model to IEEE bus systems. Moreover, authors are suggested to add more simulation results in terms of graph. This reviewer strongly recommends adding a portion of code (or at least some syntax) at the appendix of the paper. Finally, what are the technical demerits of the proposed method. 

Author Response

Comments to the Author

Response: The authors appreciate the Reviewer for reviewing this paper and providing valuable suggestions. Based on the comments, this paper has been carefully revised and improved as below.

Comment 1:Additionally, this reviewer would recommend the application of the proposed model to IEEE bus systems.

Response: The authors would like to thank the Reviewer for this comment.

The model proposed in this paper has been applied to the IEEE-39 bus system in Section 4.2 of the manuscript.

Comment 2:Moreover, authors are suggested to add more simulation results in terms of graph.

Response: The authors would like to thank the Reviewer for this comment.

According to the Reviewer’s suggestion, we have added the discussions on the method precision researches in Section 4.2 of the revised version.

All the revisions are marked in red in Section 4.2 of the revised manuscript.

Comment 3:This reviewer strongly recommends adding a portion of code (or at least some syntax) at the appendix of the paper.

Response: The authors appreciate this comment for making our paper more comprehensive.

According to the Reviewer’s suggestion, we have added the pseudo code of the proposed algorithm in the Appendix. A.

All the revisions are marked in red in Appendix. A of the revised manuscript.

Comment 4:Finally, what are the technical demerits of the proposed method.

Response: The authors would like to thank the Reviewer for this comment.

The number of units considered in this paper is only 10, which cannot be verified in a larger power system. Further improvements are needed to make it suitable for larger and more complex power systems.

All the revisions are highlighted in red in Section 5 of the revised manuscript.

Reviewer 3 Report

My Review of paper :     2273360-peer-review-v1

Title :  Research on power system day-ahead generation scheduling method considering combined operation of wind power and pumped storage power station

General notes :

  • The paper deals with an interesting topic. numerical study.
  • The Statement clarity of what is identified in the research is “good”
  • The Appropriateness of abstract as a description of the paper is “good”
  • Sufficiency of the background information presented in the introduction for better understanding the problem and the contribution is also “good”.
  • The Discussion is “acceptable”
  • Standard of English is “Good”

However, some comments need to be addressed :

1.     Please give a thorough review of the English language, all grammatical errors should be carefully eliminated.

2.     The Word format of the manuscript must be revised and performed (example: spaces between words and punctuation in general...)

3.     The original points must be explained well and Highlighted.

4.     Include the major contribution of the contributing parameters in the abstract section.

5.     Discuss the recent applications on the current study in the introduction section.

6.     There is no indication about “precision of the method” used in this work, can you give more details? Validation of the model presented in this work?

7.     Provide more detailed deep discussion of the figures presented. Extend the  discussion section with more physical significance !!.

8.     Check page 5.   Verify relations 16 and 17

9.     Check page 8; why did you presented (in table 3) only Time 11 to 15?

10.  Comparison between IEEE-4 and 289 IEEE-10 machine systems is not clear! Can you give more details?

11.  In economical  point of view, can you give more details about advantage of the technical solution proposed in this manuscript (quantitatively!)?

  1.  The conclusion section can be performed and give practical recommendations!
  2. Check References section. You must follow the template of the Journal (for some references, you don’t present pages , for example 15, 17, 21, 23)

Author Response

Comments to the Author

Response: The authors would like to thank the Reviewer for reviewing our paper and offering the insightful comments. The specific responses to the comments are given below.

Comment 1: “1.Please give a thorough review of the English language, all grammatical errors should be carefully eliminated.

Response: The authors thank the Reviewer for this comment.

According to the Reviewer’s suggestion, we have corrected the grammatical errors of this paper.

All the revisions are highlighted in red in the revised manuscript.

Comment 2:2. The Word format of the manuscript must be revised and performed (example: spaces between words and punctuation in general...)

Response: The authors thank the Reviewer for this comment.

According to the Reviewer’s suggestion, we have corrected the format errors of this paper.

All the revisions are highlighted in red in the revised manuscript.

Comment 3:The original points must be explained well and Highlighted.

Response: The authors appreciate this comment for making our paper more comprehensive.

At present, the research on the optimal operation of pumped storage power stations mainly focuses on the combined power supply with other energy sources to accept grid scheduling. The combined power supply limits the pumped storage to the power generation side, which cannot give full play to the flexibility of the pumped storage resources, and the utility of the pumped storage still has a large space for utilization. At the same time, few studies have considered the scheduling benefit caused by the loss of pumping efficiency of pumped storage power stations, and failed to consider the coordination relationship between the scheduling benefit of pumped storage power stations and the total peak shaving economy of the system and the fluctuation of new energy output. Different from existing review articles, this paper constructed the wind power-hydropower joint day-ahead dispatching model with considering the peak regulation characteristics of pumped storage. In particular, we propose a notion that regards the pumped storage power station as a technical method for mitigating the impacts of the fluctuation of wind power output.

The above revisions are marked in red in Section I of the revised manuscript.

Comment 4:Include the major contribution of the contributing parameters in the abstract section.

Response: The authors would like to thank the Reviewer for this comment.

According to the Reviewer’s suggestion, we have added the major contribution of the contributing parameters in the abstract section.

All the revisions are marked in red in Abstract Section of the revised manuscript.

Comment 5:Discuss the recent applications on the current study in the introduction section.

Response: The authors would like to appreciate Reviewer for the comment.

According to the Reviewer’s suggestion, we have added the recent applications on the current study in the introduction section.

All the revisions are marked in red in Section 1 of the revised manuscript.

Comment 6:There is no indication about “precision of the method” used in this work, can you give more details? Validation of the model presented in this work?

Response: The authors would like to appreciate Reviewer for the comment.

According to the Reviewer’s suggestion, we have added the discussions on the method precision researches in Section 4.2 of the revised version.

All the revisions are marked in red in Section 4.2 of the revised manuscript.

Comment 7:Provide more detailed deep discussion of the figures presented. Extend the discussion section with more physical significance !!.

Response: The authors would like to thank the Reviewer for the suggestion.

According to the Reviewer’s suggestion, we have added the detailed deep discussion of the figures presented in Section 4 of the revised version.

All the revisions are marked in red in Section 4 of the revised manuscript.

Comment 8:Check page 5.   Verify relations 16 and 17

Response: The authors would like to thank the Reviewer for the suggestion.

According to the Reviewer’s suggestion, we have checked the equation 16 and 17, which are the moving distance and migration angle of the colonial country. We have added the specific process in the Appendix. A.

All the revisions are marked in red in Appendix. A of the revised manuscript.

Comment 9:Check page 8; why did you presented (in table 3) only Time 11 to 15?

Response: The authors would like to thank the Reviewer for the suggestion.

The reason why the time in Table 3 starts at 11 is that the line current exceeded the limit since time 11.

Comment 10:Comparison between IEEE-4 and 289 IEEE-10 machine systems is not clear! Can you give more details?

Response: The authors would like to thank the Reviewer for the suggestion.

IEEE-4 machine systems proved robustness of the proposed method. However, IEEE-10 machine systems is used to prove effectiveness.

Comment 11:In economical point of view, can you give more details about advantage of the technical solution proposed in this manuscript (quantitatively!)?

Response: The authors would like to thank the Reviewer for the suggestion.

In the actual power system, the wind farm has a low operation cost compared with a thermal power plant, so it can be ignored, but the switching cost of the unit in the start-stop stage should be taken into account. Because the fuel characteristics of the conventional unit are affected by the load level, the power generation efficiency will be at its highest when the output of the conventional unit is kept near the rated output power. Therefore, we regard operation cost and start-stop cost as the economic indicators of this method, and we have listed the details of cost in Table 5.

All the revisions are marked in red in Table 5 of the revised manuscript.

Comment 12:The conclusion section can be performed and give practical recommendations!

Response: The authors would like to thank the Reviewer for the suggestion.

The number of units considered in this paper is only 10, which cannot be verified in a larger power system. Further improvements are needed to make it suitable for larger and more complex power systems.

All the revisions are highlighted in red in Section 5 of the revised manuscript.

Comment 13:Check References section. You must follow the template of the Journal (for some references, you don’t present pages , for example 15, 17, 21, 23)

Response: The authors apologize for the format error. It has been fixed.

Reviewer 4 Report

This article presents "Research on power system day-ahead generation scheduling method considering combined operation of wind power and pumped storage power station". An interesting article, but there are the following suggestions for their solution for publication:

The abstract of the article provides a correctly presented description of the essence of the research. 

The introduction presents the background to the research in very general terms. There is a lack of general grounding in the direction taken.  The nature of uncertainty and fluctuations and the inefficiency and instability of traditional thermal blocks are highlighted without (albeit general) argumentation in the introduction. A review of the literature in this area is lacking.

The research part of the paper presents itself well. Modelling and presentation of results clear. 

Attention is drawn to the need to complete the text under models, figures e.g. line 184, 195, 203.

The summary of the paper should be developed. The main conclusions should be highlighted and linked to practical utility. Limitations of the study and possible further recognition should be indicated. It is worth demonstrating the contribution the study makes to the literature - implications should be reinforced. 

Suggest developing the literature and reviewing it more widely.  In the background area, it is worth studying https://doi.org/10.3390/en15072470 and https://doi.org/10.3390/en12183429.

Author Response

Comments to the Author

Response: The authors appreciate the Reviewer for reviewing our paper and offering the insightful comments. The paper has been carefully revised and improved based on the comments. The specific responses to the comments are given below.

Comment 1:The introduction presents the background to the research in very general terms. There is a lack of general grounding in the direction taken.  The nature of uncertainty and fluctuations and the inefficiency and instability of traditional thermal blocks are highlighted without (albeit general) argumentation in the introduction. A review of the literature in this area is lacking.

Response: The authors would like to thank the Reviewer for the suggestion.

According to the Reviewer’s suggestion, we have added the recent applications on the current study in the introduction section.

All the revisions are marked in red in Section 1 of the revised manuscript.

Comment 2:Attention is drawn to the need to complete the text under models, figures e.g. line 184, 195, 203.

Response: The authors apologize for the format error. It has been fixed.

Comment 3:The summary of the paper should be developed. The main conclusions should be highlighted and linked to practical utility. Limitations of the study and possible further recognition should be indicated. It is worth demonstrating the contribution the study makes to the literature - implications should be reinforced.

Response: The authors would like to thank the Reviewer for the suggestion.

At present, the research on the optimal operation of pumped storage power stations mainly focuses on the combined power supply with other energy sources to accept grid scheduling. The combined power supply limits the pumped storage to the power generation side, which cannot give full play to the flexibility of the pumped storage resources, and the utility of the pumped storage still has a large space for utilization. At the same time, few studies have considered the scheduling benefit caused by the loss of pumping efficiency of pumped storage power stations, and failed to consider the coordination relationship between the scheduling benefit of pumped storage power stations and the total peak shaving economy of the system and the fluctuation of new energy output. Different from existing review articles, this paper constructed the wind power-hydropower joint day-ahead dispatching model with considering the peak regulation characteristics of pumped storage. In particular, we propose a notion that regards the pumped storage power station as a technical method for mitigating the impacts of the fluctuation of wind power output.

The above revisions are marked in red in Section I of the revised manuscript.

Round 2

Reviewer 2 Report

Thanks for the revision. This reviewer has no further questions/comments.